# Influence of Cu Content on Structure, Thermal Stability and Magnetic Properties in Fe_72−*x*_Ni_8_Nb_4_Cu*_x_*Si_2_B_14_ Alloys

**DOI:** 10.3390/ma14040726

**Published:** 2021-02-04

**Authors:** Tymon Warski, Adrian Radon, Przemyslaw Zackiewicz, Patryk Wlodarczyk, Marcin Polak, Anna Wojcik, Wojciech Maziarz, Aleksandra Kolano-Burian, Lukasz Hawelek

**Affiliations:** 1Lukasiewicz Research Network—Institute of Non-Ferrous Metals, 5 Sowinskiegostr., 44-100 Gliwice, Poland; adrianr@imn.gliwice.pl (A.R.); przemyslaw.zackiewicz@imn.gliwice.pl (P.Z.); patrykw@imn.gliwice.pl (P.W.); marcin.polak@imn.gliwice.pl (M.P.); olak@imn.gliwice.pl (A.K.-B.); 2Institute of Metallurgy and Materials Science, Polish Academy of Sciences, 25 Reymonta str., 30-059 Krakow, Poland; wojcik.a@imim.pl (A.W.); w.maziarz@imim.pl (W.M.)

**Keywords:** soft magnetic materials, metallic glass, crystallization, magnetic properties, thermal stability

## Abstract

The effect of substitution of Fe by Cu on the crystal structure and magnetic properties of Fe_72−*x*_Ni_8_Nb_4_Cu*_x_*Si_2_B_14_ alloys (*x* = 0.6, 1.1, 1.6 at.%) in the form of ribbons was investigated. The chemical composition of the materials was established on the basis of the calculated minima of thermodynamic parameters: Gibbs free energy of amorphous phase formation Δ*G^amorph^* (minimum at 0.6 at.% of Cu) and Gibbs free energy of mixing Δ*G^mix^* (minimum at 1.6 at.% of Cu). The characteristic crystallization temperatures *T_x_*_1*onset*_ and *T_x_*_1_ of the alpha-iron phase together with the activation energy *E_a_* for the as-spun samples were determined by differential scanning calorimetry (DSC) with a heating rate of 10–100 °C/min. In order to determine the optimal soft magnetic properties, the wound cores were subjected to a controlled isothermal annealing process in the temperature range of 340–640 °C for 20 min. Coercivity *Hc*, saturation induction *Bs* and core power losses at B = 1 T and frequency f = 50 Hz *P_10/50_* were determined for all samples. Moreover, for the samples with the lowest *Hc* and *P_10/50_*, the magnetic losses were determined in a wider frequency range 50 Hz–400 kHz. The real and imaginary parts of the magnetic permeability *µ*′, *µ*″ along with the cut-off frequency were determined for the samples annealed at 360, 460, and 560 °C. The best soft magnetic properties (i.e., the lowest value of *Hc* and *P_10/50_*) were observed for samples annealed at 460 °C, with *Hc* = 4.88–5.69 A/m, *Bs* = 1.18–1.24 T, *P_10/50_* = 0.072–0.084 W/kg, *µ*′ = 8350–10,630 and cutoff frequency at 8–9.3 × 10^4^ Hz. The structural study of as-spun and annealed ribbons was carried out using X-ray diffraction (XRD) and a transmission electron microscope (TEM).

## 1. Introduction

In 2019, the soft magnetic market was valued at USD 51.4 billion, with an assumed compound annual growth rate of 9.1% in the 2019–2024 period [1]. Nowadays, an increasing group of these soft magnetic materials are amorphous and nanocrystalline metal alloys, mainly based on iron, which are widely used as distribution and industrial transformers [2,3], high-efficiency photovoltaic inverters [4,5], stator cores [6] or elements of wind turbines [7,8]. For this kind of applications, materials should be characterized by magnetic properties such as low coercivity and core losses over a wide frequency range, high magnetization and permeability, but also low weight, wide operation temperature range, simple technological process and low manufacturing cost [9]. These requirements were met by FINEMET-like alloys, which have been widely used for over 20 years [10]. However, the growing demand and diversity of specialized electrical devices presents scientists and engineers with new challenges—improving or creating new materials that can improve existing technologies. This can be achieved by modifying the chemical composition and controlled heat treatment resulting from the crystallization kinetics.

One of the main disadvantages of iron-based metallic glasses is the relatively low glass forming ability (GFA; compared to Mg, Zr or Pd-based glasses) [11], which makes it difficult to obtain an amorphous structure by melt-spinning. In order to improve the GFA of the alloy, metals, and metalloids, with a negative enthalpy of the solution and a large difference in atomic size between them and the main component, Fe are introduced. However, reducing the amount of Fe in the alloy causes a significant deterioration of the magnetic properties [12]. For this reason, a compromise has to be found between GFA and the magnetic properties. In order to facilitate the planning of the chemical composition of alloys, thermodynamic calculations are used to determine the values of parameters such as Gibbs free energy of mixing (Δ*G^mix^*) or Gibbs free energy of amorphous phase formation (Δ*G^amorph^*), which are crucial for GFA [13].

The most commonly used alloys additives promoting GFA in Fe-based alloys are B and Si. It is worth noting that the influence of B on GFA is 5 times greater than that of Si [14]. However, a side-effect of using B addition is reducing the safe distance between the primary α-Fe crystallization and the secondary Fe-B crystallization and at higher concentrations it causes overlapping of these processes [15]. In order to prevent this process, Nb is used, which not only helps in the separation of crystallization peaks, but also allows grain size control [16]. It is also worth noting Ni, which is also often used to improve GFA, as well as the magnetic properties at high temperatures and Curie temperature, but reduces *Bs* and *Hc* values [17,18]. The last alloying additive worth mentioning and the effects of which will be discussed in this paper is Cu. In iron-based alloys, Cu forms nano clusters in an amorphous matrix, which during the annealing process act as a nucleation site for the crystallization of the α-Fe phase. Due to this, it is possible to obtain a fine-grained, uniform crystal structure, which results in the improvement of soft magnetic properties [19,20,21,22]. In the work of Lintao Dou et al. the effect of Cu substitution on the structure and magnetic properties in the Fe_72−*x*_Cu*_x_*B_20_Si_4_Nb_4_ bulk metallic glasses was described. It was determined that *Hc* decreased from 2.5 A/m to 1.4 A/m and the effective permeability increased from 16,200 to 24,700 for Cu = 0 at.% and Cu = 0.6 at.%, respectively. Further increase in Cu content resulted in deterioration of magnetic properties [23]. The Fe-Cu system has a positive heat of mixing equal to +13 kJ/mol [24], which should reduce the ability to form amorphous phase. However, it has been reported that the minor addition of copper can improve GFA [25,26]. A. Radoń et al. showed that the initial increase in copper content caused a reduction in Gibbs free energy of mixing Δ*G^mix^* (minimum at 0.5 and 0.75 at.% Cu content) in the Fe-Co-Mo-B-Si alloy. The Δ*G^mix^* has a minimum for 0.5 and 0.75 at.% Cu content. A further increase in copper content caused an increase in this energy [27]. We obtained similar results in our previous work on the effect of copper addition in the binary alloy Fe_76−*x*_B_14_, where the thermodynamic parameters were correlated with the crystallization kinetics, crystal structure, and magnetic properties. It was found that a small addition of copper reduces the Gibbs free energy of amorphous phase formation (Δ*G^amorph^*), which has a minimum for 0.55 at.% Cu content. Moreover, with such a copper content, the minimum value of the real and imaginary magnetic permeability was recorded and it is a “breakpoint” in the magnetic properties of the material [28]. It is worth noting, that the optimal Cu content depends on the chemical composition and the Cu effect on the properties of multicomponent amorphous and nanocrystalline iron-based alloys is not fully understood.

Fe-Ni-Nb-Si-B and Fe-Ni-Nb-Cu-Si-B systems were reported recently. Previously, Aronhime at al. presented results for (Fe_100−*x*_Ni*_x_*)_80_Nb_4_Si_2_B_14_ (30 < *x* <70) alloys, which show promising properties for applications in motors and transformers. These as-spun alloys have following magnetic properties: coercivity *Hc* = 7.0 A/m, magnetic saturation *Bs* = 1.3 T, power core loses measured at 1 T and 400 Hz *P_10/400_* = 0.9 W/kg and at 1 kHz *P_10/1000_* = 2.3 W/kg and a relatively high permeability which increases from 4000 to 16,000 after strain-annealing [29,30]. J. Gutierrez et al. studied the effect of heat treatment at 350–550 °C on magnetic and piezoelectric properties on the Fe_64_Ni_10_Nb_3_Cu_1_Si_13_B_9_ alloy. A significant improvement in *Hc* was observed from 50 A/m for as-spun to 5 A/m for annealed at 520 °C samples. Annealing at a temperature higher than 520 °C causes a slight increase in the *Hc* value. The *Bs* value ranged from 1.1 to 1.18 T [31]. The alloy Fe_67.2_Ni_10_Nb_2.9_Cu_0.8_Si_11.2_B_8_ annealed under a magnetic field called VITROPERM 250 is used industrially and reaches *Bs* = 1.24 T, 2800–4000 permeability and *Hc* < 3 A/m [32].

The present study describes the effect of Cu substitution in Fe_72−*x*_Ni_8_Nb_4_Cu*_x_*Si_2_B_14_ alloys on thermodynamic parameters, crystallization kinetics, crystal structure, and magnetic properties. Our initial chemical composition is based on the multi-component alloy Fe_72_Ni_8_Nb_4_Si_2_B_14_, which was described in the previous work [33]. The heat treatment of this material has been optimized in order to obtain the lowest possible core power losses. The best soft magnetic properties were obtained after the 20 min isothermal heat treatment at 370 °C and were characterized by *Hc* = 3.95 A/m, *P_10/50_* = 0.092 W/kg, *Bs* = 1.09 T, *µ*′ = 3100 and the cut-off frequency at 5 × 10^5^ Hz. Moreover, by performing aging process at 310–370 °C for up to 6200 min, it was possible to improve *µ*′ value up to 5000. In the present work, the Cu substitution in the alloy was optimized by thermodynamic calculations to fit the minimum Gibbs free energy of amorphous phase formation Δ*G^amorph^* (minimum at 1.6 at.% Cu) and the Gibbs free energy of mixing Δ*G^mix^* (minimum at 0.6 at.% Cu). Annealing processes were carried out to optimize the soft magnetic properties.

## 2. Materials and Methods

The master alloys of Fe_72−*x*_Ni_8_Nb_4_Cu*_x_*Si_2_B_14_ (*x* = 0.6, 1.1, 1.6) were prepared in an induction furnace under an argon atmosphere (heating at 1400–1450 °C for 15 min, casting at 1200–1260 °C) using chemical elements Fe (3N), Cu (4N), Ni (2N), and binary compound FeB_18_ (2.5N), FeNb_23_ (2.5N). The amorphous alloys in the form of a 6–7 mm width ribbons were obtained by melt spinning technique (at 30 m/s Cu wheel speed, casting at 1250–1260 °C).

In order to analyze the influence of the annealing temperature on the soft magnetic parameters the wound toroidal cores were isothermally annealed in a vacuum furnace (5·10^−3^ mbar) for 20 min at various temperatures from 340 to 640 °C at about 10 °C/min heating rate. The crystalline structure of the samples was verified by X-ray diffraction (XRD) at room temperature using a Rigaku MiniFlex 600 diffractometer (Rigaku, Tokyo, Japan) equipped with a CuKα copper tube (λ = 1.5406 Å). On the basis of differential scanning calorimetry (DSC) measurements carried out with the Netzsch DSC 214 Polyma (NETZSCH-Gerätebau GmbH, Selb, Germany) with a heating rate of 10–100 °C/min, the crystallization kinetics of amorphous materials was investigated. To determine the coercivity (*Hc*) and the magnetic saturation (*Bs*), hysteresis loops were obtain up to saturation at 50 Hz with the Remacomp C-1200 magnetic measurement system (MAGNET-PHYSIK Dr. Steingroever GmbH, Köln, Germany). The values of power losses (*Ps)* for all annealed samples were measured with a magnetic induction B = 1 T and f = 50 Hz (*P_10/50_*). Additionally for samples annealed at 460 °C the *Ps* parameters were measured in the frequency range f = 50 Hz–400 kHz and the magnetic induction B = 0.01–1 T. For samples annealed at characteristic temperatures (360, 460 and 560 °C) the complex magnetic permeability at room temperature and in the frequency range f = 10^4^–10^8^ Hz was determined with the Agilent 4294A impedance analyzer (Agilent, Santa Clara, CA, USA). Moreover, bright field (BF) and dark field (DF) images together with selected area electron diffraction patterns (SAEDPs) were recorded by using a transmission electron microscope (TEM)—Tecnai G2 F20 (200 kV) electron microscope (Thermo Fisher Scientific, Waltham, MA, USA).

## 3. Thermodynamic Calculation

To determine the optimal concentration of copper content in Fe_72−*x*_Ni_8_Nb_4_Cu*_x_*Si_2_B_14_ alloys, thermodynamic calculations were performed. Two parameters, Gibbs free energy of mixing (Δ*G^mix^*) and formation of amorphous phase (Δ*G^amorph^*), were calculated and optimal chemical compositions were chosen. First of all, the configurational entropy (*S^conf^*) was calculated according to Equation (1).
(1)$$\Delta S^{config}=-R\overset{n}{\underset{i=1}{\sum }}c_{i}lnc_{i}$$
where *R* is the gas constant and *c_i_* concentration of *i*-th chemical element.

Afterwards, enthalpies were calculated according to the simple equation using the values determined on the basis of calculations performed using Miedema’s model implemented in Miedema Calculator [34,35,36,37]. Generally, the enthalpy for the alloys can be calculated as:(2)$$\Delta H_{k}^{x}=4\Delta H_{ij}^{x}c_{i}c_{j}$$
(3)$$\Delta H^{x}=\overset{N}{\underset{k=1}{\sum }}\Delta H_{k}^{x}$$
where Δ*H^x^* is the enthalpy of mixing (Δ*H^mix^*) or formation of amorphous phase (Δ*H^amorph^*), $\Delta H_{ij}^{x}$ is the enthalpy between *i*-th and *j*-th chemical elements for equiatomic composition in a binary system (determined using a Miedema Calculator), *c_i_* and *c_j_* are the concentrations of *i*-th and *j*-th elements, *k* is the atomic pair number, *N* is the number of different atomic pairs *ij* (*N* = 15 for *n* = 6), *n* is the number of chemical elements in the alloy (in this study *n* = 6).

Taking into account this fact, that both entropy and enthalpy have a high impact on the thermodynamics of the liquid phase, and then on the crystallization process under cooling, the Gibbs free energies were calculated according to the formula:(4)$$\Delta G^{x}=\Delta H^{x}-T\Delta S^{conf}$$
where: Δ*G^x^* is the Gibbs free energy of mixing (Δ*G^mix^*) or formation of amorphous phase (Δ*G^amorph^*), and *T* is the average melting temperature.

The analysis results are presented in Figure 1. The first minimum was related to the lowest value of Gibbs free energy of formation of amorphous phase, whereas the second one to the lowest value of Gibbs free energy of mixing. The value of the Δ*G^amorph^* firstly rapidly decreases with increasing copper content, which is related to the positive contribution of configurational entropy. After reaching the minimum at 0.6 at.%. the negative impact of Cu can be observed and this is related to the contribution from Δ*H^amorph^*. A similar relationship can be observed for Δ*G^mix^,* however, the minimum was observed for much higher Cu content equal to 1.6 at.%. According to that, the Fe_71.4_Ni_8_Nb_4_Cu_0.6_Si_2_B_14_ and Fe_70.4_Ni_8_Nb_4_Cu_1.6_Si_2_B_14_ alloys were chosen from the optimization process to the analysis of structure and magnetic properties in these studies. Additionally, to broaden the analysis a third Fe_70.9_Ni_8_Nb_4_Cu_1.1_Si_2_B_14_ alloy was selected, with Δ*G^amorph^* higher than calculated minimum and Δ*G^mix^* lower than calculated minimum.

## 4. Results

### 4.1. Kinetic and Structure Studies of As-Spun Alloys

In order to confirm the structure of the as-spun materials, an X-ray diffraction study was carried out. The XRD patterns were presented in Figure 2. The 1st and 2nd order of the amorphous halo are clearly visible confirming amorphousness of the materials.

α-Fe crystallization processes were recorded by differential scanning calorimetry (DSC). Samples were heated up to 580 °C with heating rates in the range from 10 to 100 °C/min. The DSC signals with marked temperatures of onset (*T_x_*_1*onset*_) and crystallization peak (*T_x_*_1_) of α-Fe are presented in Figure 3a–c. At lower heating rates the *T_x_*_1*onset*_ temperatures are very similar for all samples. However, as the heating rate increased the differences in characteristic temperatures are more visible—the higher the Cu content, the lower the *T_x_*_1*onset*_ temperature. The influence of Cu addition on crystallization temperature (*T_x_*_1_) is not clearly noticeable, therefore the average activation energy was calculated. For this purpose the Kissinger model for not-isothermal crystallization was used [38]:(5)$$ln\left(\frac{\beta }{T^{2\;}}\right)=-\frac{E_{a}}{RT}+C_{1}$$
where: *β* is the heating rate, $E_{a}$ is the activation energy, *R* is the gas constant, *T* is the temperature of maximum of crystallization peak and $C_{1}\;$ is the constant. The slope of linearly fitted $ln\left(\beta /T^{2\;}\right)$ vs. 1/T curves is used to obtain $E_{a}$.

The Kissinger plots $ln\left(\beta /T^{2\;}\right)$ vs. 1/T with the determined activation energies $E_{a}$ are presented in Figure 3d–f. For the sample with 0.6 at.% of Cu the average $E_{a}$ is equal to 312.02 kJ/mol. For samples with a higher content of Cu = 1.1 at.% and 1.6 at.% the average $E_{a}$ decreases and is equal to 305.27 and 297.97 kJ/mol, respectively. There is a correlation between the decrease in $E_{a}$ and the onset of *T_x1_* crystallization temperatures with the increase in the Cu content. The Cu atoms agglomerate and act as a nucleation site for the α-Fe phase at the Cu nanocluster and amorphous phase boundary, which facilitates the precipitation of the α-Fe phase.

### 4.2. Magnetic Properties Studies

The dependence of the coercivity *Hc*, the magnetic saturation *Bs* and the core power losses at 1 T and 50 Hz *P_10/50_* on the annealing temperatures for the as-spun and annealed alloys was presented in Figure 4. A significant improvement in the soft magnetic properties was noticed after the heat treatment at 340 °C. Similar values of *Hc*, *Bs*, and *P_10/50_* have been maintained up to 540 °C for the material with Cu content equal to 1.6 at.% and up to 580 °C for the materials with 0.6 at.% and 1.1 at.% of Cu. Heat treatment at higher temperatures was associated with the deterioration of the soft magnetic properties, which was related to the separation of the boride phase. It is worth noting that the temperature of the boride phase formation decreases with increasing copper content in the materials. Table 1 shows magnetic properties of ribbon cores before and after 20 min isothermal annealing at characteristic temperatures: 360, 460, and 560 °C. Additionally, the results for Fe_72_Ni_8_Nb_4_Si_2_B_14_ annealed at 370 °C are provided for comparison. The alloys annealed at 360 °C differ significantly from the as-spun alloys. There was a 3.36–3.42-fold decrease in *Hc*, a 4.16–4.63-fold decrease in *P_10/50_*, and an increase in *Bs* by 7.4–18.6% compared to the non-annealed samples. The improvement of soft magnetic properties is related to the relaxation of stresses that arose during the rapid cooling of materials in the melt-spinning process. The temperature of 460 °C corresponds to the optimal annealing temperature for which the *Hc* value is the lowest for all ribbons. It is worth mentioning, that the material with 1.1 at.% Cu content had the lowest *Hc* and *P_10/50_* values. The last characteristic temperature, i.e., 560 °C, corresponds to the annealing temperature, at which the boride phase begins to form. At this annealing temperature, the materials have the highest value of *Bs* equal to 1.31 T for 0.6 at.% Cu and 1.41 T for both 1.1 and 1.6 at.% Cu.

One of the most important parameters in terms of applicability are the core power losses *Ps*. Figure 5 shows the *Ps* parameter of metallic ribbons annealed at 460 °C depending on the magnetic induction for different frequencies. The *Ps* characteristics are summarized in Table 2. Additionally, the results of the previous work were added to the table for comparison (Fe_72_Ni_8_Nb_4_Si_2_B_14_ annealed at 370 °C). Initially, the value of the *Ps* parameter at lower frequencies is similar for all samples. However, as the frequency increases, the difference becomes apparent (*Ps* values for Cu content 1.6 < 0.6 < 1.1). The sample with Cu = 1.1 at.% has the highest values of *Ps*. For the sample with Cu = 1.6 at.% (minimum Δ*G^mix^*), with increasing frequency, the *Ps* value is lower by 5–25% than for samples with Cu = 1.1 at.%. For the sample with Cu = 0.6 at.% (minimum Δ*G^amorph^*), initially the losses are higher by 4% (*P_10/100_*), and then they decrease up to 15% (*P_0.5/400,000_*). It is worth mentioning that at low frequencies the values of the *Ps* parameter have a different relationship than at high frequencies (*P_10/50_* values for Cu content 1.1 < 0.6 < 1.6).

The magnetic permeability *µ*′ and magnetic permeability loss *µ*″ in the function of frequency for samples annealed at 360, 460, and 560 °C were presented in Figure 6 and comparison of characteristic parameters is summarized in Table 3. Additionally, the results for Fe_72_Ni_8_Nb_4_Si_2_B_14_ annealed at 370 °C are provided. Among the samples annealed at 360 °C, the highest magnetic permeability *µ*′value has the sample with the Cu = 1.6 at.% (*µ*′ = 4623) and the lowest with Cu = 0.6 at.% (*µ*′ = 2152). After increasing the annealing temperature to 460 °C, the values of the real part of magnetic permeability increased drastically more than 2.2–4.5 times and the cut-off frequency is shifted towards lower frequencies. The highest value of *µ*′ among all measured samples, equal to 10,630, has a sample with a Cu content of 1.6 at.%, while the next one with a copper content of 0.6% at. has the value 9252. It is worth noting that both materials have a minimum Gibbs free energy value both the mixing (minimum Δ*G^mix^*) and the amorphous phase formation (minimum Δ*G^amorph^*). At the annealing temperature of 560 °C, materials with copper content Cu = 0.6 at.% and 1.1 at.% retained similar properties to their counterparts annealed at a lower temperature. However, for the sample Cu = 1.6 at.%, the permeability decreased significantly to 3679 and the cut-off frequency increased to 3.35 × 10^5^ Hz, which may be related to the different state of the crystal structure.

### 4.3. Crystal Structure of Annealed Alloys

To determine the influence of the structure on the magnetic properties, XRD and TEM studies were performed for samples annealed at characteristic temperatures. Figure 7 shows the XRD patterns of samples annealed at 360, 460, and 560 °C. The alloys after annealing at 360 °C are still in an amorphous state as a relaxed glass. It was noticed that with increasing temperature, reflections corresponding to the α-Fe phase begin to appear and at 560 °C the α-Fe peaks have higher intensities and become narrower.

Figure 8 presents a set of bright field (BF) images along with selected area electron diffraction patterns (SAEDPs) of Fe_72−*x*_Ni_8_Nb_4_Cu*_x_*Si_2_B_14_ ribbons containing 0.6 at.%, 1.1 at.% and 1.6 at.% Cu, annealed at 360, 460, and 560 °C. Regardless of the Cu content, ribbons annealed at 360 °C show a fully amorphous structure with the characteristic halo observed in SAEDPs. Ribbons heat treated at 460 °C were two-phase composed of α-Fe crystals embedded in an amorphous matrix while the samples after annealing at 560 °C were fully crystalized. SAEDPs were identified as α-Fe phase. However, in SAEDPs of ribbons containing 1.6 at.% of Cu annealed at 560 °C additional reflections corresponding to the Fe_2_B phase were found. Based on dark field (DF) images (an example of the set of BF, SAEDPs and DF images obtained from the marked reflections of α-Fe phase of Fe_72−*x*_Ni_8_Nb_4_Cu*_x_*Si_2_B_14_ ribbon heat treated at 460 °C), the average α-Fe crystallite sizes were calculated with standard deviations for each sample. The values are presented in Figure 8. With an increase in Cu content the size of the crystallites decreases slightly while with increasing temperature it increases. The smallest α-Fe crystallites, 8.0 ± 2.6 nm in size, were found in the Fe_70.4_Ni_8_Nb_4_Cu_1.6_Si_2_B_14_ ribbon, annealed at 460 °C.

The Figure 9 presents high-resolution transmission electron microscopy (HRTEM), fast Fourier transform (FFT) and inverse fast Fourier transform (IFFT) images of Fe_70.4_Ni_8_Nb_4_Cu_1.6_Si_2_B_14_ annealed at 560 °C. In the FFT, reflections corresponding to the α-Fe crystals and the additional phase were found. The extra phase reflections were indexed according to the Fe_2_B phase with a crystallite size below 2 nm.

## 5. Conclusions

The influence of copper addition on thermodynamic parameters, crystallization kinetics, crystal structure, and magnetic properties of Fe_72−*x*_Ni_8_Nb_4_Cu*_x_*Si_2_B_14_ alloys was investigated. Thermodynamic studies shows two minima of Gibbs free energy, i.e., Δ*G^amorph^* for Cu = 0.6 at.% and Δ*G^mix^* for Cu = 1.6 at.%. The study on the crystallization kinetics shows that with the increase in Cu content, the crystallization onset temperature *T_x_*_1*onset*_ and activation energy *E_a_* of the α-Fe crystallization process decrease. However, the difference in *T_x_*_1_ crystallization temperature between the materials is small. The annealing process has been optimized to determine the temperature corresponding to the lowest values of *Hc* and *P_10/50_*. A significant improvement in the soft magnetic properties was noted after annealing at 360 °C. Increasing the annealing temperature to 560 °C did not cause significant changes in the *Hc* and *P_10/50_* values. The samples annealed at 460 °C had minimal *Hc* and *P_10/50_* values. The best soft magnetic properties: *Hc* = 4.88 A/m, *Bs* = 1.24 T, *P_10/50_* = 0.072 W/kg, have been found for a sample with Cu = 1.1 at.%. More in-depth studies of the core power losses show a linear increase in *Ps* values as a function of frequency. Magnetic permeability studies have proven that the measurement materials, after annealing at 460 °C, can reach *µ*′ values in the range of 8350–10,630, with a cut-off frequency of nearly 10^5^ Hz. It is interesting to note that the highest values of *µ*′ correspond to materials with 1.6 at.% and 0.6 at.% Cu content (minimum of Gibbs free energy). It is also worth paying attention to the positive effect of the Cu addition on the magnetic properties in comparison with Cu-free counterpart (Fe_72_Ni_8_Nb_4_Si_2_B_14_ alloy) studied in our previous work [33]. The materials obtained in this work have higher *Bs* values and lower *Ps* values in the entire tested frequency and magnetic induction range. These improved magnetic parameters (higher *Bs* and lower *Ps* in the broad frequency range) are crucial especially in electric motor application. Structure studies confirmed the amorphous state of the as-spun materials and relaxed glass for samples annealed at 360 °C. These studies also allowed to determine the microstructure evolution caused by the annealing process. It was observed that the addition of Cu had a significant effect on the fine-grained structure, reducing the size of the crystallites of the alloys annealed at 460 °C (from 10.4 nm for Cu = 0.6 at.% to 8.0 nm for Cu = 1.6 at.%). Moreover, increasing the Cu content reduces the amount of crystallites present in the amorphous matrix after the annealing process. However, it has negative effect on thermal stability and lowers the crystallization temperature of the Fe_2_B phase. In summary, the high value of magnetic permeability *µ*′, low core power losses *Ps* and coercivity *Hc*, satisfactory *Bs* and good thermal stability of the discussed alloys make it possible to find the application as replacements of existing materials used in the electrical devices and power industry.

## Figures and Tables

**Figure 1 materials-14-00726-f001:**
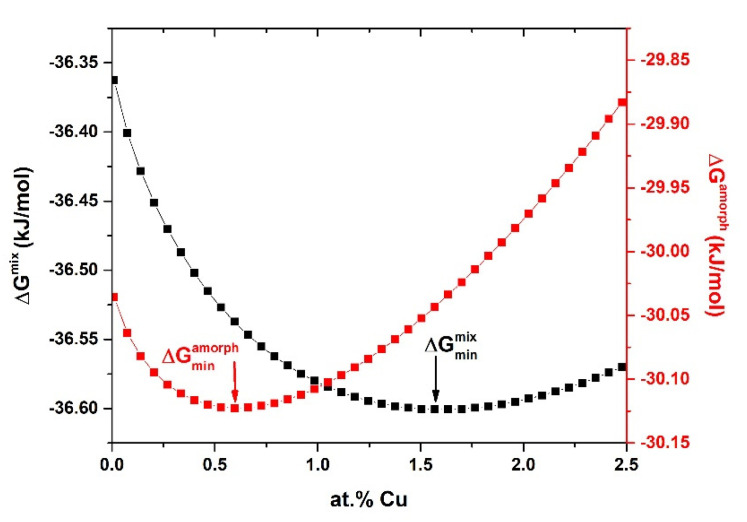
The influence of the copper content on the Gibbs free energy of mixing (Δ*G^mix^*) and formation of amorphous phase (Δ*G^amorph^*).

**Figure 2 materials-14-00726-f002:**
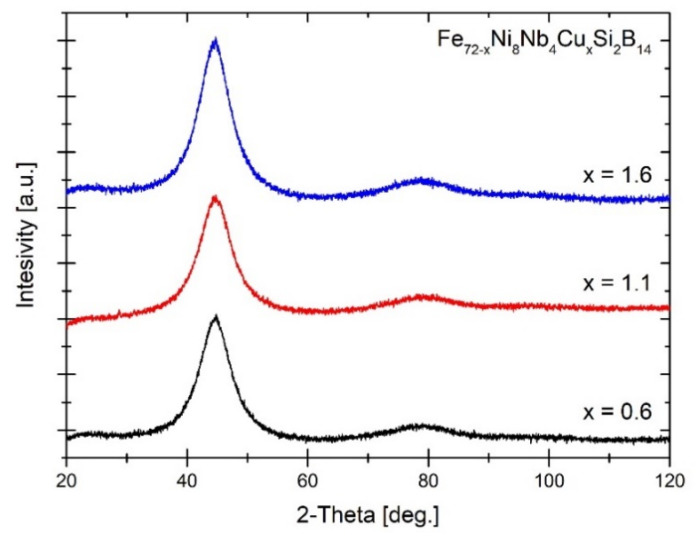
The XRD patterns of as-spun alloys.

**Figure 3 materials-14-00726-f003:**
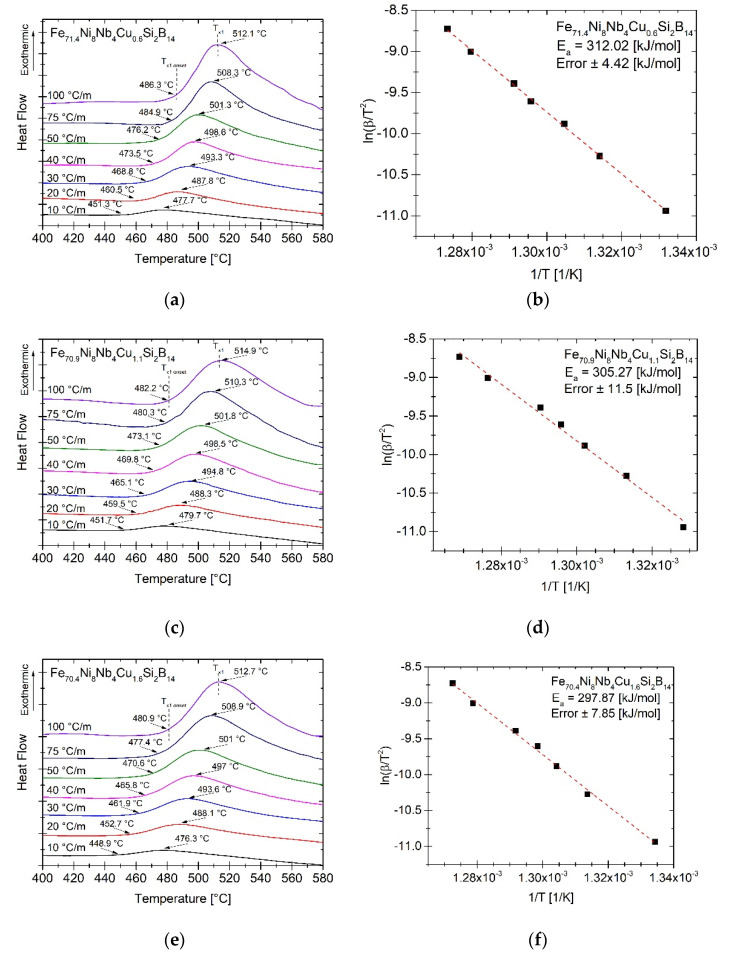
The DSC thermograms for heating rates (**a**,**c**,**e**) ranged from 10 to 100 °C/min and Kissinger plots determined for crystallization peak temperatures T_x1_ (**b**,**d**,**f**) of as-spun amorphous alloys.

**Figure 4 materials-14-00726-f004:**
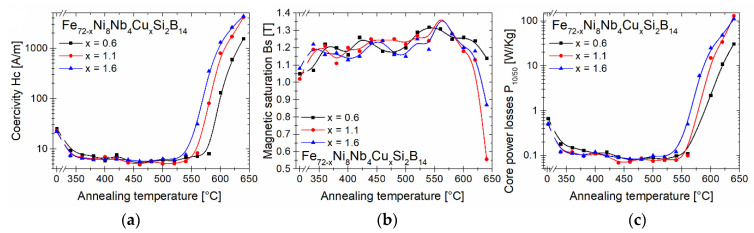
Coercivity *Hc* (**a**), magnetic saturation *Bs* (**b**) and core power losses *P_10/50_* (**c**) of measured cores.

**Figure 5 materials-14-00726-f005:**
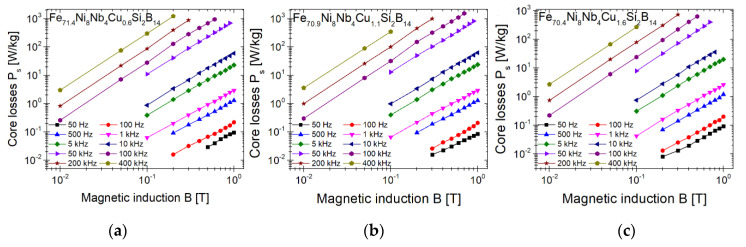
Core power losses *Ps* of annealed at 460 °C metallic ribbons: Fe_71.4_Ni_8_Nb_4_Cu_0.6_Si_2_B_14_ (**a**), Fe_70.9_Ni_8_Nb_4_Cu_1.1_Si_2_B_14_ (**b**) and Fe_70.4_Ni_8_Nb_4_Cu_1.6_Si_2_B_14_ (**c**) measured at 50 Hz–400 kHz in function of magnetic induction.

**Figure 6 materials-14-00726-f006:**
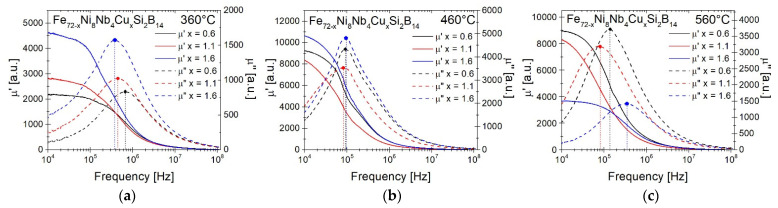
Magnetic permeability *µ*′ and magnetic loss permeability *µ*″ dependence as a function of frequency for samples annealed at 360 °C (**a**), 460 °C (**b**), and 560 °C (**c**).

**Figure 7 materials-14-00726-f007:**
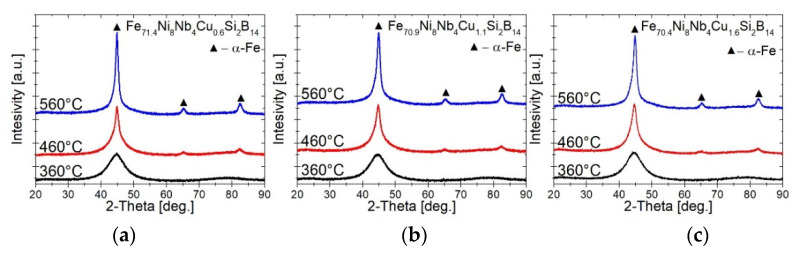
XRD patterns of Fe_71.4_Ni_8_Nb_4_Cu_0.6_Si_2_B_14_ (**a**), Fe_70.9_Ni_8_Nb_4_Cu_1.1_Si_2_B_14_ (**b**) and Fe_70.4_Ni_8_Nb_4_Cu_1.6_Si_2_B_14_ (**c**) annealed at 360 °C, 460 °C, and 560 °C.

**Figure 8 materials-14-00726-f008:**
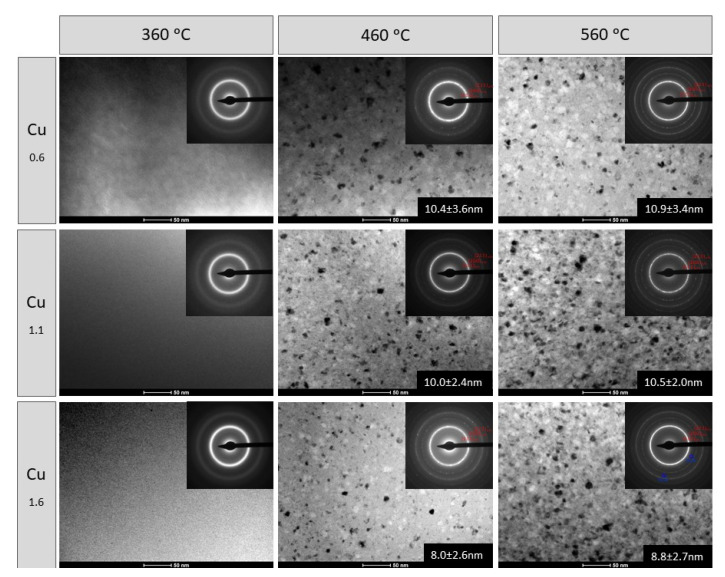
Microstructural evolution of Fe_72−*x*_Ni_8_Nb_4_Cu*_x_*Si_2_B_14_.

**Figure 9 materials-14-00726-f009:**
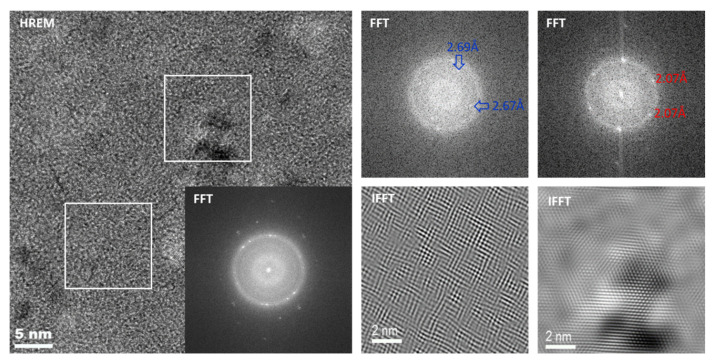
HRTEM, FFT, and IFFT images of Fe_70.4_Ni_8_Nb_4_Cu_1.6_Si_2_B_14_ annealed at 560 °C.

**Table 1 materials-14-00726-t001:** Coercivity *Hc*, magnetic saturation *Bs* and core power losses *P_10/50_* of amorphous ribbon cores as-spun and annealed at characteristic temperatures.

Alloy	Annealing Temperatures [°C]	*Hc* [A/m]	*Bs* [T]	*P_10/50_* [W/kg]
**Fe_71.4_Ni_8_Nb_4_Cu_0.6_Si_2_B_14_**	As-spun	25.7	1.05	0.67
	360	7.64	1.22	0.15
	460	5.17	1.18	0.079
	560	7.19	1.31	0.11
**Fe_70.9_Ni_8_Nb_4_Cu_1.1_Si_2_B_14_**	As-spun	23.3	1.02	0.5
	360	6.8	1.21	0.12
	460	4.88	1.24	0.072
	560	8.28	1.41	0.1
**Fe_70.4_Ni_8_Nb_4_Cu_1.6_Si_2_B_14_**	As-spun	22.2	1.08	0.51
	360	6.48	1.16	0.11
	460	5.69	1.24	0.084
	560	31.4	1.41	0.5
**Fe_72_Ni_8_Nb_4_Si_2_B_14_** [33]	370	3.95	1.09	0.092

**Table 2 materials-14-00726-t002:** Different core power losses values of Fe_72−*x*_Ni_8_Nb_4_Cu*_x_*Si_2_B_14_ alloys (*x* = 0, 0.6, 1.1, 1.6).

*Ps* [W/kg]	*x* = 0.6	*x* = 1.1	*x* = 1.6	*x* = 0 [33]
***P_10/100_***	0.22	0.21	0.2	0.28
***P_10/500_***	1.3	1.3	1.2	1.8
***P_10/1000_***	2.9	2.9	2.6	4.2
***P_10/5000_***	23	24	20	35
***P_5/10,000_***	18	19	16	29
***P_5/50,000_***	232	261	202	321
***P_1/100,000_***	28	32	24	39
***P_1/200,000_***	87	102	79	110
***P*** *_**0.5/400,000**_*	76	89	67	88

**Table 3 materials-14-00726-t003:** Magnetic permeability *µ*′ at 10^4^ Hz and cut-off frequency for annealed toroidal cores at characteristic temperatures.

Alloy	Annealing Temperature [°C]	Magnetic Permeability *µ*′ at 10^4^ Hz	Cut-Off Frequency [Hz]
**Fe_71.4_Ni_8_Nb_4_Cu_0.6_Si_2_B_14_**	360	2152	6.7 × 10^5^
	460	9252	8.9 × 10^4^
	560	8946	1.39 × 10^5^
**Fe_70.9_Ni_8_Nb_4_Cu_1.1_Si_2_B_14_**	360	2825	4.17 × 10^5^
	460	8350	8 × 10^4^
	560	8309	6.8 × 10^4^
**Fe_70.4_Ni_8_Nb_4_Cu_1.6_Si_2_B_14_**	360	4623	3.8 × 10^5^
	460	10,630	9.3 × 10^4^
	560	3679	3.35 × 10^5^
**Fe_72_Ni_8_Nb_4_Si_2_B_14_** [33]	370	3100	5.07 × 10^5^

## Data Availability

The data presented in this study are available on request from the corresponding authors.

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
