# Peer review of "Influence of Cu Content on Structure, Thermal Stability and Magnetic Properties in Fe72−xNi8Nb4CuxSi2B14 Alloys"

_materials, 2021, doi:10.3390/ma14040726_

Round 1

Reviewer 1 Report

In this paper the authors studied the possibility to replace Fe by Cu on the crystal structure and magnetic properties of 11 Fe72-xNi8Nb4CuxSi2B14 alloys in the form of ribbons. The effects of this replacing on thermodynamic parameters, crystallization kinetics, crystal structure and magnetic properties were extensively investigated. The paper is well organized, clear with a satisfactory state of novelty. This referee suggests the publication on materials after the following main revisions:
1. It is suggested to reduce the abstract extention;
2. Check some typos, i.e. ‘’_[17–19].’’ Al line 67 in pp. 2;
3. The title axes of Figure 6 are difficult to read, check it please.

Author Response

Thank you very much for your comment and suggestions. In response to them:

  1. The abstract has been slightly shortened.
  2. The text has been corrected
  3. Axis titles have been changed to be easier to read.

Reviewer 2 Report

The manuscript describes the structural and magnetic properties of a series of multi-component alloys Fe72-xNi8Nb4CuxSi2B14, for x = 0.6, 1.1, 1.6. The materials were prepared in the form of ribbons by the melt-spinning technique and subjected to controlled annealing. The experiment was designed to clarify the effect of partial substitution of Fe with Cu on the properties of the similarly obtained parent unsubstituted alloy Fe72Ni8Nb4Si2B14 (x = 0) examined previously by nearly all members of the present team.

As a whole, the paper is reasonably well written and the level of new results is acceptable. From my point of view, a few complements and improvements should be added, however.

1. English and misprints

  • Line 168: missing verb
  • Line 181: One reads Figure 1. The influence on the copper content on the Gibbs free energy of mixing......??. 
  • Lines 186-187 missing definite article 
  • Line 193 and Line 313: incorrect tense
  • Lines 213 - 214 erroneously read: The dependence of the annealing temperatures on the coercivity Hc, the magnetic  saturation Bs and the core power losses at 1 T and 50 Hz P10/50 for the as-spun and annealed alloys was presented in Figure 4.
  • Line 216: maintained not maitained
  • Lines 219-220: boron phase or boride phase?
  • Lines 145 and 286-287: SAEDPs not SADEPs 
  • Line 288: In Figure 8 not in the Figure 8 .....
  • Line 298: The values are presented in Figure 8 but not in the Figure 8
  • Line 304: the abbreviations HREM, FFT, and IFFT are to be introduced 
  • Lines 305-306 read: In the FFT the reflections corresponding to α-Fe crystals and an additional phase were found. They were indexed in accordance to Fe2B phase with the size less than 2 nm. Please edit these sentences as well as the preceding text correctly in English, clarifying all the findings and the corresponding dimensions of the inclusions in the amorphous matrix.
  • Line 313: shown ?
  • 4.3 Crystal Strucure:misprint

Comment: The manuscript contains a mixture of sentences with an illogically used sequence of tenses. Please, find and rewrite the corresponding texts.

For example, one reads: The XRD patterns were presented in Figure 2. The 1st and 2nd order halo are clearly visible, which confirms amorphousness of the materials. 

2. From the perspective of potential technological applications, it would be interesting to comment on the final magnetic results obtained

3. A substantial drawback of the present version of the manuscript is the lack in Table 1, Table 2, and Table 3 of data for the corresponding magnetic parameters of the parent alloy with x=0. To a large extent, this compromises by essence the depth of Part 4. Conclusions.

Author Response

Thank you very much for your comment and suggestions. In response to them:

  1. The text has been corrected
  2. Information on the applicability of the materials in electric motors due to their magnetic properties has been added to the conclusions.
  3. The results, for the material with Cu = 0 at. % described in the previous work, have been added to the tables and magnetic properties of this material have been compared with properties of the materials from current work.

Reviewer 3 Report

Dear authors,

Thank you for submitting the manuscript entitled 'Influence of Cu content on structure, thermal stability and magnetic properties in Fe72-xNi8Nb4CuxSi2B14 alloys'. I recommend to publish it in its present form.

The manuscript concerns with a very interesting and modern class of metallic magnetic materials - glasses. The study on influence of stoichiometric changes and thermal treatment on magnetism and crystallization behavior is realized with a convincing combination of chemical, metallurgical, theoretical and physical methods. This leads to comrehensible insights into the behavior of materials excluded from standard structure-property relation considerations due to their amorphous nature. The results of this broad study are compared to the available literature phases and a nicely written Introduction provides a more general access for the reader to this modern and special field of interesting materials. The Conclusion nicely wraps up the bottom-line results and points out its significance to future applications. The list of references is concise. I have no urgent need for additions or alterations.

Sincere greetings

Author Response

Thank you very much for your comment.